# Clinical Diagnosis and Treatment of Leigh Syndrome Based on *SURF1*: Genotype and Phenotype

**DOI:** 10.3390/antiox10121950

**Published:** 2021-12-05

**Authors:** Inn-Chi Lee, Kuo-Liang Chiang

**Affiliations:** 1Division of Pediatric Neurology, Department of Pediatrics, Chung Shan Medical University Hospital, Taichung 40201, Taiwan; 2Institute of Medicine, School of Medicine, Chung Shan Medical University, Taichung 40201, Taiwan; 3Department of Pediatric Neurology, Kuang-Tien General Hospital, Taichung 43303, Taiwan; lambier.tw@yahoo.com.tw

**Keywords:** mitochondrial disorders, complex IV assembly, Leigh syndrome

## Abstract

SURF1 encodes the assembly factor for maintaining the antioxidant of cytochrome c oxidase (COX) stability in the human electron respiratory chain. Mutations in SURF1 can cause Leigh syndrome (LS), a subacute neurodegenerative encephalopathy, characterized by early onset (infancy), grave prognosis, and predominant symptoms presenting in the basal ganglia, thalamus, brainstem, cerebellum, and peripheral nerves. To date, more than sixty different *SURF1* mutations have been found to cause SURF1-associated LS; however, the relationship between genotype and phenotype is still unclear. Most SURF1-associated LS courses present as typical LS and cause early mortality (before the age of ten years). However, 10% of the cases present with atypical courses with milder symptoms and increased life expectancy. One reason for this inconsistency may be due to specific duplications or mutations close to the C-terminus of the SURF1 protein appearing to cause less protein decay. Furthermore, the treatment for SURF1-associated LS is unsatisfactory. A ketogenic diet is most often prescribed and has proven to be effective. Supplementing with coenzyme Q and other cofactors is also a common treatment option; however, the results are inconsistent. Importantly, anti-epileptic drugs such as valproate—which cause mitochondrial dysfunction—should be avoided in patients with SURF1-associated LS presenting with seizures.

## 1. Genetic Background of Leigh Disease

Leigh disease, also called Leigh syndrome (LS), is a genetically heterogeneous disease and can be (1) maternally inherited through mutations in mitochondrial DNA (mtDNA) encoding complex I (*MTND1, MTND2, MTND3, MTND4, MTND5,* and *MTND6*), complex IV (*MTCO3*), complex V (*MTATP6*) and mitochondrial translation (*MTTI, MTTK, MTTL1, MTTV,* and *MTTW)*; (2) sex-linked *PDHA1* causing pyruvate dehydrogenase deficiency; or (3) autosomal recessive due to mutations in nuclear-encoded complex subunits complex (*SDHA* and *SDHAF1* in complex II deficiency; *UQCRQ, BCS1L,* and *TTC19* in complex III deficiency; and *NDUFA4, SURF1, COX10, COX15, SCO2, PET100, LRPPRC, TACO1,* and *ETHE1* in complex IV deficiency), autosomal recessive in nuclear genes *(PDHB, PDHX, DLAT, DLD, LIPT1, LIAS, TPK1, SLC19A3,* and *SLC25A19)* causing pyruvate dehydrogenase deficiency [1,2], and autosomal recessive in complex assembly genes [3,4,5,6]. To date, more than 75 causative genes have been identified to be involved in the biochemical pathways in LS [3,4,5,6,7].

## 2. Cytochrome c Oxidase Deficiency and LS

Cytochrome c oxidase (COX) (ferrocytochrome c:oxygen oxidoreductase, EC 1.9.3.1) is the terminal component of the mitochondrial respiratory chain that catalyzes the transfer of electrons from reduced cytochrome c to oxygen. It is composed of thirteen subunits, ten of which are encoded by nuclear genes and three of which are encoded by the mitochondrial genome. These protein subunits are assembled by at least six nuclear-encoded assembly factors: *SURF1* [MIM 185620], *SCO1* [MIM 603644], *SCO2* [MIM 604272], *COX10* [MIM 602125], *COX15* [MIM 603646], and *LRPPRC* [MIM 607544) [6,8,9,10,11]. COX deficiency (MIM 220110) is a major cause of LS (MIM 256000) [4,6,8,9]. Mutations in COX assembly genes are the major causes of isolated COX deficiency.

## 3. Human *SURF1* Gene and SURF1 Protein

The human surfeit 1 (*SURF1*) gene encodes a three-hundred amino acid mitochondrial protein necessary for the assembly and maintenance of the COX holoenzyme which is essential for energy production in the human body [5,6,8,9,11,12,13]. The *SURF1* gene is located on chromosome 9p34, in a region of clustering of surfeit genes, where the genomic structure is well-conserved from chicken to human. The protein Surf1 contains two transmembrane domains—one at the N-terminus and the other at the C-terminus—which are essential for the function of the protein [12,13,14,15,16]. The function of the human *SURF1* gene is still not fully verified [17,18,19]. Nevertheless, the number of *SURF1* mutations is increasing, and more than forty mutations have been reported in Leigh or Leigh-like disease (Figure 1) [4,6,12,13,14,15,16,17,20,21,22,23,24,25,26,27,28,29,30,31,32,33,34,35,36,37,38]. As next-generation sequencing approaches have been widely applied to the clinical diagnosis of LS, a correlation between the LS phenotype and genotype is now possible. Here, we review the literature concerning *SURF1*-associated Leigh syndrome, and the phenotype and genotype of *SURF1*-associated Leigh syndrome in children. Considering the rarity of *SURF1*-associated Leigh syndrome, its’ poor outcomes with early mortality, and its’ difficulty to treat, we review in detail the genotype and phenotype, the typical and atypical courses of Leigh syndrome, and treatment.

## 4. Typical LS Course

Most LS patients present with early-onset (since infancy) subacute neurodegenerative encephalopathy and have a grave prognosis. LS is characterized by bilateral symmetrical necrotic lesions in the basal ganglia, midbrain, thalamus, and spinal cord [6,34,37,38]. A typical course of LS includes symptoms of neurodevelopmental regression, brainstem and basal ganglia signs, lactic acidosis, and unique magnetic resonance imaging (MRI) findings. Other presentations include: failure to thrive, microcephaly, hypertrichosis, and myopathy [19,31,34,38,39]. LS is a heterogeneous disorder that can be classified into typical and atypical courses. First described by Rahman et al. [4], this reference is still widely used as the diagnostic criteria for LS. Patients with LS and *SURF1* mutations were thought to manifest the typical course, presenting as early neurodevelopmental regression, showing typical neuroradiological features, and usually dying before ten years of age [13,14,15,30,33]. The atypical course has rarely been mentioned in the literature, with reports [33,40] citing patients with LS with *SURF1* mutations that have survived for more than ten years in one patient.

## 5. Typical and Atypical LS Classification

Typical LS was first defined by Rahman et al. [4] with a strict set of criteria, including unique MRI findings. However, Rahman et al. [4] described a *SURF1*-associated LS patient with leukodystrophy and suggested that the phenotype associated with SURF1 protein deficiency should include leukodystrophy. Leukodystrophies are a large heterogeneous group of neuroradiological features affecting the white matter of the central nervous system [40]. The leukodystrophy in this patient [19] showed extensive involvement in the white cerebral matter. There are some reports that *SURF1*-associated LS patients may have unique MRI findings rather than basal ganglia involvement (Table 1). 

In general, LS patients with *SURF1* mutations belong to the typical LS group. According to a study by Tiranti et al. [13], of the twenty-four patients classified with typical LS according to Rahman criteria, 17 had a *SURF1* mutation, grave prognosis and died before they were ten years old. In another study [41], five out of twenty-one *SURF1*-associated LS patients had an atypical course, with one patient surviving for over twenty years. In one study of 152 patients with leucodystrophies, next-generation genetic analysis revealed one individual harboring *SUR1F* mutations [40]. The author explained that a mutation close to the C-terminus could lead to a milder course. A boy, who showed an almost normal MRI finding at the age of three, presented with severe isolated COX deficiency at the age of ten, and a follow-up MRI showed cerebellar atrophy and cystic-like changes inferior to the pontine part of the brain stem [11]. Another patient [23] with a c.C688T mutation and a heterozygous 15 bp tandem duplication (820_820) had a more favorable course and survived for ten years, which seemed to indicate that in-frame duplication did not result in the complete loss of the function of SURF1 and that it conferred a better prognosis. The reasons for the patient’s long survival are unclear; it might have been due to a compensatory mechanism that existed when the *SURF1* mutation occurred. A gene defect cannot be completely expressed in clinical phenotypes. Whether there are compensatory mechanisms in the human body when *SURF1* mutations occur is worthy of further investigation. There is one report [35] of LS patients with the same genotype presenting with extremely different phenotypes. Identical mutations can be associated with different phenotypes and varied clinical and MRI findings [Wolf, 1997 #53]. Genetic heterogeneity includes mutations at different gene loci or allelic mutations within a single gene, resulting in a similar phenotype. Epigenetic phenomena provide a plausible explanation. The relationship between pathogenic mutations and disease phenotypes is becoming increasingly complex.

## 6. The Possible Mechanism of the *SURF1* Gene and SURF1 Protein

LS was first reported in 1977, 29 years after the first case of COX deficiency was reported [39], in a patient who was documented to have a c.370G>A *SURF1* gene mutation [22]. Extensive reports have been published on *SURF1* mutations in LS, but the exact role of SURF1 protein has not been fully verified. In a murine model, *SURF1*^−/−^ mice had lower birth weights but reached body weights comparable to those of their siblings; however, they displayed a mild motor delay. Mild elevation of lactic acid in tissue was also found, analogous to that observed in humans with a *SURF1* mutation [Bartke, 2008 #54]. Interestingly, *SURF1* knockout mice were more resistant to neuronal damage induced by kainic acid and glutamate. In a zebrafish model, morpholino-induced COX deficiency inhibited the expression of COX and SURF1, which resulted in a developmental defect in cardiac function and swimming behavior. In addition, apoptosis was dramatically increased in the hindbrain and neural tube, and secondary motor neurons were either abnormal or absent, which explains the motility defect on the swimming behavior [42]. These two animal studies imply that COX and SURF1 are important for animal brain development and learning. It is well known that without the SURF1 protein contribution to the assembly of COX, the energy production system of mitochondria will not be functional. 

## 7. MRI Findings in Typical LS and the *SURF1* Mutation

MRI findings in the *SURF1* mutation of LS were rarely mentioned before the 21st century and, in the few reports mentioning them, no detailed description was provided [30,33,37]. Since MRI is a key tool for diagnosing LS, we summarized the MRI findings from published case reports in Table 2. Most cases of *SURF1-mutated* LS also involve changes to the structure of the basal ganglia (subthalamic nuclei, caudate nuclei, substantia nigra, tegmentum, and periaqueductal gray matter) or structures of the rhombencephalon-like dentate nucleus of the cerebellum and medulla [16,21,23,24,29,36,37], although there is some confusion regarding MRI findings. Using the Rahman criteria [4], most typical MRI findings in LS with *SURF1* mutations involved the basal ganglia (Table 2), but some reports found that other areas (Table 2), especially the subthalamic area and the substantia nigra, were involved. From the perspective of neuroanatomy, these two areas may be considered part of the basal ganglia, although this is not universally accepted. The probable reason for the sensitivity to cytochrome c deficiency in these locations is the presence of the same highly oxygen-dependent neurotransmitter. Isolated leukodystrophy is unusual in patients with LS and *SURF1* mutations, therefore other mitochondrial diseases, such as COX10 gene mutations and metabolic disorders, should be considered for diagnosis [9,26].

## 8. Peripheral Neuropathy and Myopathy in LS

Some patients with *SURF1*-associated LS were diagnosed with muscle weakness or muscle-biopsy-proven myopathy [41]. Patients with LS and *SURF1* deficits had findings of decreased cytochrome c oxidase staining and a defect in myelination in muscle and nerve biopsies, respectively [41,42,43]. Strikingly, most patients had c.845_846delCT mutations in one or two alleles. All these patients had a specific phenotype in their muscle biopsy findings, and all were originally from Eastern Europe. This implies that the c.845_846delCT mutation is more likely to be detected in Slovenia than elsewhere. A muscle biopsy may be interpreted as normal if the histochemical panel does not include a COX reaction, but patients still have some vague muscle symptoms such as muscle weakness or floppy baby syndrome (hypotonia). A muscle biopsy with a COX histochemical reaction is a more precise diagnosis. Peripheral neuropathy has rarely been reported in patients with LS, but in some cases, it does occur. Peripheral neuropathy is not listed as one of the symptoms in the original description of the disease by Leigh [38] or in the inclusive criteria by Rahman et al. [4]; however, neither Leigh nor Rahman et al. have documented nerve biopsies or spinal cord MRIs. One possible reason for the presentation of peripheral neuropathy may be related to spinal cord involvement in LS and demyelination of peripheral nerves. Histological patterns can be very informative in mitochondrial diseases but are not specific with respect to the mutated genes.

## 9. Genotypes from the Literature and Our Patients with LS and *SURF1* Mutations

Although LS can be genetically inherited through mutations in the mtDNA and in the *PDHA1, SCO1, SCO2*, and *COX10* genes, studies [13,34,44] next-generation sequencing studies have shown that *SURF1* is an important gene for LS. Further, mtDNA mutations account for 25% of LS cases [45], whereas most patients have nuclear DNA mutations [46]. In one study, 8 (8.3%) of 96 LS cases in Northern Europe were reported to carry *SURF1* mutations [47]. In a Japanese study, 7 (10%) of 70 cases with a genetic diagnosis of LS had *SURF1* mutations [48]. We have summarized more than sixty different mutations in *SURF1* (Figure 1). The most common mutation was c.845_c.846delCT in exon 8, which was detected in all patients from Slovenia, the Czech Republic, and Poland [25,36,43], and accounts for approximately 25% of all *SURF1* mutant alleles. The second is c.312_c.321del10insAT in exon 4, which accounted for about 17% of all mutations commonly found in Western, Eastern, and Northern Europeans. The third is c.C688T, which is most commonly found in Northern Europeans. We found c.653_c.654delCT in two reports [41,49] suggesting that this mutation probably originated in East Asia. One study [50] of 124 cases of LS in China showed that only 4.8% of their patients had mitochondrial DNA mutations, but approximately 20.2% had *SURF1* mutations: c.G604C and c.653_c.654delCT. We concluded that different ethnicities have different mutations. It seems that c.845_c.846delCT is typical for people of Slovenian descent as a founder effect [43].

## 10. Missense and Nonsense Distributions

Based on the missense and nonsense mutations, the ratio of nonsense to missense mutations was 15: 11 (Figure 1). It is thus postulated [41,51] that nonsense mutations contribute to more severely nonfunctional proteins that cause a more severe phenotype. Among *SURF1* mutations, the majority were splice-site, frameshift, and nonsense mutations (Figure 1). Most *SURF1* mutations occur in exons 6, 7, and 8, which account for approximately half of the total mutations. Strangely enough, the length of exons 6 to 8 accounts for only a small proportion of the total *SURF1* gene. The longest exon, exon 5, contains 192 base pairs but harbors a small number of different mutations (Figure 1).

## 11. Correlation of Phenotype and Genotype

No clear associations between genotype and phenotype were identified in the clinical features, biochemical analyses, and MRI findings of the mutations. This agrees with other studies, [19,20,25] and it is difficult to predict LS with *SURF1* mutations by genotype because patients with the same mutation/s will often have extremely different expressions. Individual clinical features, such as accelerated neuroregression, should be an important but unpredictable prognostic factor. MRI is also an important, but not an absolute, prognostic factor. In a study from Turkey, 16 cases from 14 families were reported to harbor a recurrent point mutation, c.769G>A [52]. The author used three-dimensional (3D) structure prediction for two novel missense variants (C.595_597delGGA and c.356C>T), which cause protein degradation and are predicted as likely pathogenic based on conservation. In this case series, the outcome was poor in all cases, except for one patient with compound heterozygous mutations of c.595_597delGGA (p.Gly199del) and c.751+1 G>A who survived until 18 years of age. However, the reason for this remains unclear. In this case series, no epilepsy was reported, but there were more movement disorders (dystonia and ataxia). Patients with hypotonicity showed shorter survival compared to patients without hypotonicity. However, genotype and phenotype were not related [52]. In one study, patients with *SURF1* mutations had worse outcomes of LS compared to those in patients with LS caused by mt-ATP6, *NDUF, SLC19A3,* and *SUCLA2* mutations [47]. For LS caused by *SURF1* mutations, the outcomes cannot be predicted based on the results of genetic studies and studies on cytochrome c activity. The possible compensatory mechanisms in patients with *SURF1* mutations are worthy of further study. In a previous study, the COX enzyme assembled in the absence of Shy1p (the yeast homolog of SURF1) appears to be structurally and enzymatically normal. In vitro labeling studies additionally indicated that mitochondrial translation is significantly increased in the shy1 null mutant strain, reflecting a compensatory mechanism for reduced respiratory capacity [18]. Increased mitochondrial copy number could be a mechanism of compensation and suggests a new direction for mitochondrial biogenesis [53].

## 12. Therapeutic Approaches

There is currently no curative therapy for SURF1-associated LS. Managing mitochondrial disease is largely supportive. In many cases, the treatment outcome is not promising for *SURF1*-associated LS since treatments are similar to those used for other mitochondrial diseases encoded by mutations in other genes. *SURF1*-associated mutations predispose patients to symptoms in the brain, peripheral nerves, and muscles, yet symptoms in the liver and heart are not typical, and only approximately 30% of patients present with seizures [41]. To manage symptoms of the nervous system, understanding the complex interactions between metallothionein (MT) redox biology, bioenergetics, and cellular signaling may provide some more efficacious therapeutic strategies [54]. Currently, a ketogenic diet is the most prescribed treatment and has proven to be effective. Coenzyme Q and other cofactors are commonly used as treatments, but their effects are inconsistent. Antiepileptic drugs such as valproate, which causes mitochondrial dysfunction, should be avoided for the treatment of seizures in patients with *SURF1*-associated LS.

## 13. Management and Medications 

### 13.1. Ketogenic Diet and High-Fat Diet

The ketogenic diet is one of the oldest, yet most promising, therapeutic approaches for mitochondrial defects. Ketones decrease oxidative stress and increase antioxidants and the scavenging of free radicals. Thus, the ketogenic diet can be adopted to help manage mitochondrial disorders. The diet is composed of four-parts fat for every one-part carbohydrate or protein consumed, therefore energy metabolism bypasses the glycolytic pathway [55]. In early 1987, a male patient with LS due to lack of PDHc adopted the ketogenic diet [56,57]. Subsequently, his lactic acid and pyruvate plasma levels decreased, and his neurological symptoms temporarily improved. Potential benefits of the ketogenic diet were observed under specific conditions in various models. However, the patient’s condition gradually deteriorated and he ultimately died. On the contrary, a case series study reported that the ketogenic diet could extend longevity and lessen mental problems in patients with PDHc deficiency [58]. Therefore, in the treatment of LS caused by PDHc deficiency, a ketogenic diet is often used and the results are promising. A ketogenic diet can also treat intractable seizures with mitochondrial disorders. The purpose of a ketogenic diet is to replace glucose with fat as an energy resource. Ketone bodies can also be used to treat seizures. One study [59] involving 14 children, reported fewer seizures after being prescribed a ketogenic diet. A high-fat diet constituting 50–60% of daily calories can increase electron transfer, which is similar to a ketogenic diet. A high-fat diet improves the outcomes of mitochondrial disease [59,60]. The ketogenic diet has proven beneficial in select patients with mitochondrial respiratory chain defects [59]. The available experimental evidence indicates that the ketogenic diet can positively affect mitochondrial bioenergetics, mitochondrial reactive oxygen species (ROS)/(reduction–oxidation) redox metabolism, and mitochondrial dynamics [61,62,63]. In one study, four of twenty cases of ketogenic diet reversed cardiomyopathy and movement disorders. In five of these cases involving mitochondrial DNA deletion, myopathy rhabdomyolysis led to the cessation of the ketogenic diet. Three patients with *POLG* mutations died while being on ketogenic diets; however, this was not different compared to individuals with *POLG* mutations without ketogenic diets [63,64]. When considering a ketogenic diet for mitochondrial disorders, the high rate of adverse effects should be taken into account along with the spectacular improvements that have occurred in individual cases, particularly for young children.

### 13.2. Non-Invasive Positive Pressure Ventilation (NIPPV)

For children with LS and brain stem dysfunction, adequate ventilation support with NIPPV is recommended if they also have apnea. NIPPV is the optimal choice in most cases when the patient needs home care [55].

### 13.3. Drugs that Should Be Avoided

Potentially, mitochondrion-toxic drugs should be avoided, particularly for seizures. Prescribing valproate for patients with polymerase r gene defects is contraindicated because it can cause acute liver damage, and ultimately the patient will require a liver transplant. In fact, valproate should be avoided in all mitochondrial diseases.

### 13.4. Coenzyme Q (CoQ) and Vitamin Treatment

The drugs used for LS treatment include coenzyme Q10, L-carnitine, alfa-lipoic acid, creatine monohydrate, biotin, thiamine, and riboflavin. Decades of research have shown that vitamins and cofactors have no significant effect on respiratory chain disease; however, these are still commonly used to treat mitochondrial disease [65]. Coenzyme Q10 is the most commonly recommended therapy for mitochondrial diseases. Coenzyme Q10 derivatives include idebenone, decylubiquinone, and duroquinone. Quinones can transfer electrons from complex I to complex II and from complex II to complex III in the respiratory chain, leading to increased ATP production, and stabilizing membrane calcium channels. The most common side effects are gastrointestinal problems. Coenzyme Q10 supplementation can restore electron flow and improve clinical symptoms associated with coenzyme Q10 deficiency [57]. However, apart from coenzyme Q10 deficiency, this supplementation has limited benefits for other mitochondrial diseases. However, supplementation with EPI-743 (a derivative of coenzyme Q10) in some individual cases of LS due to *SURF1* mutation can prevent the progression of the disease and improve the quality of life and motor function score [66,67]. Numerous open-label clinical trials of coenzyme Q10 have been performed with varying success. Results from over 50 clinical trials of coenzyme Q10 suggest a marginal but real treatment effect [68]. To improve coenzyme Q10 the bioavailability, the idebenone truncated side-chain analog was synthesized over a decade ago and repurposed to treat inherited mitochondrial disease [69,70]. Commonly, the effects were not consistent except in the case of some forms of primary ubiquinone deficiency, which showed great improvement after ubiquinone supplementation [63]. The standard palliative dose is 5 mg/kg [54] to 30 mg/kg for more severe mitochondrial disorders. 

Vitamin supplementation is an alternative way to improve the neurological symptoms of LS. Thiamine is an important cofactor involved in energy metabolism in brain tissue. Thiamine is a cofactor of the pyruvate dehydrogenase complex. Thiamine deficient LS is caused by mutations in the *SLC19A3* gene, which encodes a thiamine transporter. The early administration of thiamine and biotin has a significant and immediate therapeutic effect [62] on biotin-responsive basal ganglia [67,71]. If untreated, it can lead to severe neurological impairment and death. However, except for LS caused by variants of the *SLC19A3* gene, thiamine has no lasting effect on LS caused by most other causes. A recent case study reported the discovery of a novel mutation in p.P298L *SURF1* in thiamine responsive LS and revealed compromised cytochrome c oxidase activity [72,73]. Idebenone, dimethylglycine, vitamin E, vitamin C, and triacetyluridine have been studied; however, they are only palliative and the results are inconsistent [54]. Vitamin cocktails in patients diagnosed with mitochondrial disease can be considered. Retinoic acid and vitamin E can be given to patients [74,75]. A high dose of biotin (10–20 mg/kg) and thiamine (100–300 mg) can be administered to treat mitochondrial disorders [75]. 

Riboflavin treatment reportedly improved LS patients displaying neurological symptoms due to PDHc deficiency [71,76]. High doses of riboflavin can increase muscle strength and reduce lactic acidosis in complex I deficiency caused by the *ACAD9* mutation [73,77]. *ACAD9* is a key factor in maintaining complex I function. If it is defective, it will affect the operation of the mitochondrial oxidative phosphorylation system and ATP production. High-dose riboflavin is a powerful antidote, but it is not a universal solution to other mitochondrial respiratory chain defects. Folinic acid is recommended as a supplement to improve Kearns-Sayre syndrome with secondary folate deficiency [77,78], based on the hypothesis that the failure of mitochondrial ATP production cannot support the active transport of folic acid across the blood-brain barrier. Other mitochondrial encephalopathies may also have secondary folate deficiency [78,79]. Nevertheless, the above conclusions are based on a small number of case reports. Other nutraceuticals have been proposed for the treatment of LS or other mitochondrial disorders. These include L-carnitine, α-lipoic acid, and creatine-monohydrate [79,80].

### 13.5. Mitochondrial Biogenesis and Gene Replacement Therapy

In LS with *SURF1* mutations, mitochondrial biogenesis is impaired; thus, augmenting mitochondrial biogenesis could be a potential therapeutic approach. Recent reports indicate melatonin promotes mitochondrial biogenesis in the treatment of Alzheimer’s disease and Parkinson’s disease [53,81] and could be a potential drug owing to its influence on mitochondrial physiology. Gene therapy for spinal muscle atrophy (SMA) has recently been developed and has shown promising effects [82]. In LS with *SURF1* mutations, intrathecal delivery of adeno-associated viral vector serotype 9 (AAV9)/human SURF1 (h*SURF1*) was studied in mice and was found effective in improving the biochemical abnormalities induced by SURF1 deficiency, thus showing potential applicability for patients with *SURF1*-related Leigh syndrome in the future [83]. 

## 14. Outcomes

Treatment for LS is unsatisfactory. The majority of patients with LS will have poor neurodevelopmental outcomes and even die before they turn ten years old. In a Japan study from 124 mitochondrial genetic confirmed patients, seven were SURF mutations. Of those, three need mechanical ventilation, and two need tube feeding [12,48,84]. In principle, a genetic diagnosis is mandatory and can be performed by a mitochondrial panel and whole exon sequencing. Understanding genetic defects and the relationship between genotype and phenotype should be emphasized. It is, however, rare for patients to live for more than twenty years [41]. Managing mitochondrial disease is largely supportive and palliative, and its effects are still not promising.

## 15. Conclusions

*SURF1* mutations present as typical or atypical LS, indicating disease involved in different brain tissues. The same genotype can present with different MRI images in patients. No clear phenotype or genotype was found for *SURF1* mutations. The majority of *SURF1*-associated LS cases present as typical LS and cause early mortality before 10 years of age. However, atypical courses account for 10% of the cases, which present with milder symptoms in patients who survive longer. One reason for this is that duplication and mutation cause less protein decay close to the C-terminus of the SURF1 protein. The treatment for *SURF1*-associated LS is unsatisfactory. A ketogenic diet is most commonly used and has proven to be effective. However. Coenzyme Q and other cofactors are commonly used, but their effects have been inconsistent. For seizures in patients with *SURF1*-associated LS, antiepileptic drugs, such as valproate, that cause mitochondrial dysfunction should be avoided.

When patients present with Leigh syndrome or leukodystrophy, *SURF1* should be considered. Coenzyme Q and vitamin cocktails can be supplied early, despite the relatively poor outcomes. It is essential to investigate new therapeutic approaches, such as improving mitochondrial biogenesis, to complement the classical treatments that generally have low efficacy. The development of gene replacement therapy has potential in clinical practice for treating LS with *SURF1* mutations in the future. 

## Figures and Tables

**Figure 1 antioxidants-10-01950-f001:**
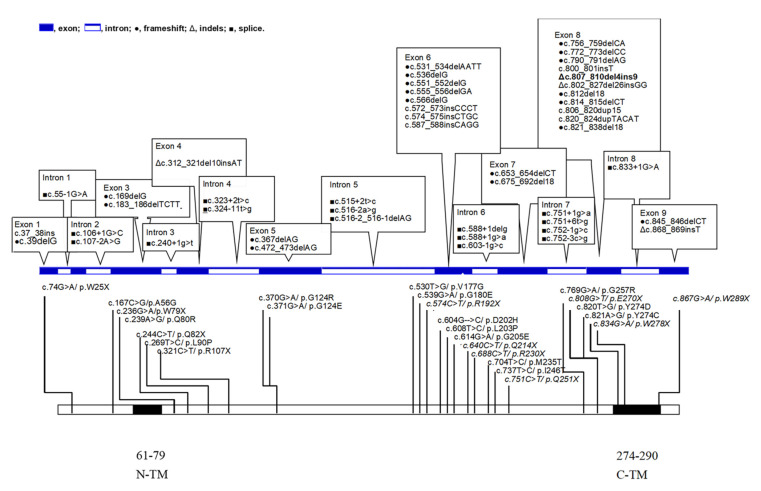
Distribution of *SURF1* mutations based on published literature.

**Table 1 antioxidants-10-01950-t001:** Atypical Leigh syndrome with *SURF1* mutations as defined by Rahman’s criteria based on published literature.

DNA Change (Allele 1/Allele 2)	Age(y) (Dx/Alive/ Died/Gender)	Phenotype	Matched Criteria	Lactate/ Pyruvate	CT/MRI	Ethnicity	COX Activity	Reference
c.867G>A/ c.G867G>A	2/died 5/M	Early motor delay, hypotonia, ataxia, tongue fasciculation, hypertrichosis, apnea with sudden death at 5 years old	3/4	+/NA	Caudal part of medulla, cerebellum	German	<30%	[34]
c.312del10insAT/ c.312del10insAT	3/died 9/M	Mental retardation, ataxia, cannot walk, ophthalmoplegia, central apnea, died due to diazepam injection at 9 years old	2/4	+/NA	CT: normal	Danish	5–18%	[24]
c.312-321delinsAT/ c.572_573insCCCT	3.5/alive 10/M	MR, brain stem signs, tube feeding, wheelchair dependency	2/4	−/−	mild MRI change	NA	NA	[11]
c.790-791delAG/ c.790-791delAG	2/alive 2/F	Early motor delay, failure to thrive, microcephaly, short stature, regression, hypertrichosis	2/4	+/+	Leukodystrophy, dentate nucleus., medulla	NA	<30%	[19]
c.618G>C/ c.751C>T	2/died 7.8/F	Hypotonia, tremor, ataxia and deafness, brain stem signs, and progressive	2/4	−/NA	Leukodystrophy	French	NA	[27]
c.688C>T/ c.751+1G>A	0.4/died 2.1/M	Feeding difficulties, short stature, muscle weakness, nystagmus	3/4	+	Autopsy at 5 years old, brain stem	NA	<30%	[23]
c.653_54delCT/ c.807_810del4ins9	22/alive 22/F	Ataxia, ophthalmoparesis, hearing loss, short stature	2/4	−	Basal ganglia	Asian	Decreased	[41]
c.169delG/ c.530T>G	1.8/alive 1.8/M	Developmental delay, muscle weakness, hypotonia, seizures, ptosis, dysmorphism, episodic coma	2/4	+/+	Leukodystrophy	Hispanic	NA	[41]
c.324-11T>G/ c.324-11T>G	2/alive 2/F	Developmental regression, failure to thrive, microcephaly	2/4	−/−	Leukodystrophy	Asian	NA	[41]
c.167C> G/ c.751+6T>C	10/alive 10/M	Mental retardation, ataxia, dystonia chorea, intractable seizures	2/4	−/+	Basal ganglia, leukodystrophy	Caucasian	32%	[41]
c.688C>T/ c.806_820dup	1.3/alive 10/M	Feeding intolerance, short stature, muscle weakness since 1 year, ophthalmoplegia, dystonia, choreoathetosis, ataxia, hirsutism, alive at 10 years old	3/4	+/NA		Swedish	<25%	[23]

Dx indicates diagnosis; M, male; F, female; COX, cytochromome c oxidase; CT, computed topography; MRI; magnetic resonance imaging; NA, non-available.

**Table 2 antioxidants-10-01950-t002:** Magnetic resonance imaging findings of the 13 published cases of Leigh syndrome.

DNA Change (Allele 1/Allele 2)	Age (y) Dx/Alive/Died	Caudate Nulceus (62%)	Subthalamic Nulceus (54%)	Substantia Nigra (62%)	Tecmentum (46%)	Dentate Nulceus (46%)	Cerebellar White Matter (38%)	Medulla/S. Cord (62%)	Cerebral White Matter (8%)	Referrence
c.244C>T/ c.244C>T	2/alive 2		**+**	**+**	**+**			**+/−**	**+**	[37]
c.530T>G/ c.530T>G	1.9/alive 3		**+**	**+**				**+**		[37]
c.312del10insAT/ c.688C>T	1.5/alive 1.5	**+**				**+**	**+**	**+**		[17]
c.312del10insAT/ c.845delCT	NA/alive 2.6	**+**	**+**	**+**	**+**	**+**		**+**		[24]
c.312del10insAT/ c.688C>T	NA/alive 0.8	**+**								[24]
c.312del10insAT/ c.820_824dupTACAT	NA/alive 0.8	**+**								[16]
c.240 + 1G>T/ c.531_534del AAAT	NA/died 1.8		**+**	**+**	**+**		**+**	**+**		[29]
c.320del10insAT/ c.320 del10insAt	1.3/alive 6.7	**+**		**+**	**+**	**+**	**+**	**+**		[23]
c.566delG	1/alive 3		**+**	**+**		**+**	**+**	**+/+**		[29]
c.772_773delCC/ c.772_773delCC	1/alive 4	**+**	**+**	**+**	**+**	**+**	**+**	**+**		[29]
c.240 + 1G>T/ c.531_534delAAAT	1.3/died 1.8		**+**	**+**						[21]
c.320del10insAT/ c.812del18	1/died 2.8	**+**								[36]
c.790delAG/ c.820T>G	0.8/alive 1.6	**+**			**+**	**+**				[33]

NA, not available; Y, year; nu, nucleus; del, deletion; ins, insertion; dup, duplication.

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
