# Peer review of "Clinical Diagnosis and Treatment of Leigh Syndrome Based on SURF1: Genotype and Phenotype"

_antioxidants, 2021, doi:10.3390/antiox10121950_

Round 1

Reviewer 1 Report

Lee et al report on “Clinical Diagnosis and Treatment of Leigh Syndrome based on 2 SURF1: Genotype and Phenotype”.

I have only some major comments and questions to do.

Please, update reference 11, that is very old and I am not sure if authors can state that “Mutations in the nuclear-en-42 coded COX structural/functional subunits have not been reported” (see for example Frazier et al (JBC 2019)

Line 55: Please, rewrite this sentence: “Only recently has clinical testing been available for SURF1; thus, 55 a correlation between the LS phenotype and genotype is now possible”.

Line 83:  Please, consider to change this sentence: “Leukodystrophies are a large heterogeneous group of genetic diseases affecting the white matter of the central nervous system”. Leukodystrophies are not a group of genetic diseases, but a neuroradiological feature than can be cause by either genetic or environmental conditions.

Line 155: I do not believe that “Patients with LS and SURF1 deficits have characteristic patterns in muscle biopsies [42-44]” as the authors claim. Histological patterns in mitochondrial disease can be very informative, but are not specific regarding the mutated genes. In other words, COX negative fibers for example, is a common finding in either several nuclear and mitochondrial DNA mutations. The same applies for this statement “All these patients had a specific phenotype in their muscle biopsy findings (line 157).

Line 170: Please, update the following paragraph. “Although LS can be genetically inherited by mutations on the PDHA1, SCO1, SCO2 171 and COX10 genes, studies [14, 35, 45] show that approximately one-third of patients with  LS carry SURF1 mutations, and that the ratio does not seem to differ by ethnicity”. The number of mutated genes leading to LS has increasing in the last decade and include other that those stated by the authors. Once again, reference are old and need to be updated.  

Line 186: Correlation of phenotype and genotype should be further detailed, especially the last part of the paragraph “We do not suggest that outcomes can be predicted based on the results of genetic studies and studies on cytochrome c activity in patients with LS and SURF1 mutations. Possible compensation mechanisms in patients with SURF1 mutations are worthy of further study.” Which mechanisms do the authors mean?. Probably, with the application of machine learning techniques, especially when NGS massive data are available, could in part elucidate this correlation.

Conclusions (line 337): Authors state that “Coenzyme Q and vitamin cocktails can be supplied early, despite the relatively poor outcomes”. Owing the lack of evidence regarding this issue, I would conclude that it is mandatory to investigate new therapeutic approaches (improve mitochondrial biogenesis, for example, and others) instead to maintain the classical in general non.-effective treatments).

Author Response

Reviewer 1

We are grateful for the opportunity to improve our manuscript and we thank the editorial board and the reviewers for their thoughtful and helpful comments and criticisms. We have modified the paper as suggested. Following are our point-by-point responses. We have also highlighted the principal changes in Microsoft Word in the revised text.

  1. I have only some major comments and questions to do.

Please, update reference 11, that is very old and I am not sure if authors can state that “Mutations in the nuclear-en-42 coded COX structural/functional subunits have not been reported” (see for example Frazier et al (JBC 2019)

Reply: We have rewritten this section, changed to:

“2. Cytochrome c oxidase deficiency and LS

Cytochrome c oxidase (COX) (ferrocytochrome c:oxygen oxidoreductase, EC 1.9.3.1) is the terminal component of the mitochondrial respiratory chain that catalyzes the transfer of electrons from reduced cytochrome c to oxygen. It is composed of thirteen subunits, ten of which are encoded by nuclear genes and three of which are encoded by the mitochondrial genome. These protein subunits are assembled by at least six nuclear encoded assembly factors: SURF1 [MIM 185620], SCO1 [MIM 603644], SCO2 [MIM 604272], COX10 [MIM 602125], COX15 [MIM 603646], and LRPPRC [MIM 607544) [6, 8-11]. COX deficiency (MIM 220110) is a major cause of LS (MIM 256000) [4, 6, 8, 9]. Mutations in COX assembly genes are the major causes of isolated COX deficiency.”

  1. Line 55: Please, rewrite this sentence: “Only recently has clinical testing been available for SURF1; thus, 55 a correlation between the LS phenotype and genotype is now possible”.

Reply: We have rewritten this sentence of the section, and changed to:

“3. Human SURF1 gene and SURF1 protein

The human surfeit 1 (SURF1) gene encodes a three-hundred amino acid mitochondrial protein necessary for the assembly and maintenance of the COX holoenzyme which is essential for energy production in the human body [5, 6, 8, 9, 11-13].  The SURF1 gene is located on chromosome 9p34, in a region of clustering of surfeit genes, where the genomic structure is well-conserved from chicken to human. The protein Surf1 contains two transmembrane domains—one at the N-terminus and the other at the C-terminus—which are essential for the function of the protein [12-16]. The function of the human SURF1 gene is still not fully verified [17-19]. Nevertheless, the number of SURF1 mutations is increasing, and more than forty mutations have been reported in Leigh or Leigh-like disease (Figure 1) [4, 6, 12-17, 20-38]. As next-generation sequencing approaches have been widely applied to the clinical diagnosis of LS, a correlation between the LS phenotype and genotype is now possible. Here, we review the literature concerning SURF1-associated Leigh syndrome, and the phenotype and genotype of SURF1-associated Leigh syndrome in children. Considering the rarity of SURF1-associated Leigh syndrome, its’ poor outcomes with early mortality, and its’ difficulty to treat, we review in detail the genotype and phenotype, the typical and atypical courses of Leigh syndrome, and treatment.”

  1. Line 83:  Please, consider to change this sentence: “Leukodystrophies are a large heterogeneous group of genetic diseases affecting the white matter of the central nervous system”. Leukodystrophies are not a group of genetic diseases, but a neuroradiological feature than can be cause by either genetic or environmental conditions.

Reply: We have corrected the sentence in the section, and changed to:

“5. Typical and atypical LS classification

Typical LS was first defined by Rahman et al. [4] with a strict set of criteria, including unique MRI findings. However, Rahman et al. [4] described a SURF1-associated LS patient with leukodystrophy and suggested that the phenotype associated with SURF1 protein deficiency should include leukodystrophy. Leukodystrophies are a large heterogeneous group of neuroradiological features affecting the white matter of the central nervous system [40]. The leukodystrophy in this patient [19] showed extensive involvement in the white cerebral matter. There are some reports that SURF1-associated LS patients may have unique MRI findings rather than basal ganglia involvement (Table 1).”

  1. Line 155: I do not believe that “Patients with LS and SURF1 deficits have characteristic patterns in muscle biopsies [42-44]” as the authors claim. Histological patterns in mitochondrial disease can be very informative, but are not specific regarding the mutated genes. In other words, COX negative fibers for example, is a common finding in either several nuclear and mitochondrial DNA mutations. The same applies for this statement “All these patients had a specific phenotype in their muscle biopsy findings (line 157).

Reply: We have rewritten the sentence in the section, changed to:

“8. Peripheral neuropathy and myopathy in LS

Some patients with SURF1-associated LS were diagnosed with muscle weakness or muscle-biopsy-proven myopathy [41]. Patients with LS and SURF1 deficits had findings of decreased cytochrome c oxidase staining and a defect in myelination in muscle and nerve biopsies, respectively [41-43]. Strikingly, most patients had c.845_846delCT mutations in one or two alleles. All these patients had a specific phenotype in their muscle biopsy findings, and all were originally from Eastern Europe. This implies that the c.845_846delCT mutation is more likely to be detected in Slovenia than elsewhere. A muscle biopsy may be interpreted as normal if the histochemical panel does not include a COX reaction, but patients still have some vague muscle symptoms such as muscle weakness or floppy baby syndrome (hypotonia). A muscle biopsy with a COX histochemical reaction is a more precise diagnosis. Peripheral neuropathy has rarely been reported in patients with LS, but in some cases it does occur. Peripheral neuropathy is not listed as one of the symptoms in the original description of the disease by Leigh [38] or in the inclusive criteria by Rahman et al. [4]; however, neither Leigh nor Rahman et al. have documented nerve biopsies or spinal cord MRIs. One possible reason for the presentation of peripheral neuropathy may be related to spinal cord involvement in LS and demyelination of peripheral nerves. Histological patterns can be very informative in mitochondrial diseases, but are not specific with respect to the mutated genes.”

  1. Line 170: Please, update the following paragraph. “Although LS can be genetically inherited by mutations on the PDHA1, SCO1, SCO2 171 and COX10 genes, studies [14, 35, 45] show that approximately one-third of patients with  LS carry SURF1 mutations, and that the ratio does not seem to differ by ethnicity”. The number of mutated genes leading to LS has increasing in the last decade and include other that those stated by the authors. Once again, reference are old and need to be updated.  

Reply: We have rewritten the sentence in the section, and changed to:

“9. Genotypes from the literature and our patients with LS and SURF1 mutations

Although LS can be genetically inherited through mutations in the mtDNA and in the  PDHA1, SCO1, SCO2, and COX10 genes, studies [13, 34, 44] next-generation sequencing studies have shown that SURF1 is an important gene for LS. Further, mtDNA mutations account for 25% of LS cases [45], whereas most patients have nuclear DNA mutations [46]. In one study, 8 (8.3%) of 96 LS cases in Northern Europe were reported to carry SURF1 mutations[47]. In a Japanese study, 7 (10%) of 70 cases with a genetic diagnosis of LS had SURF1 mutations[48]. We have summarized more than sixty different mutations in SURF1 (Figure 1). The most common mutation was c.845_c.846delCT in exon 8, which was detected in all patients from Slovenia, Czech Republic and Poland [25, 36, 43], and accounts for approximately 25% of all SURF1 mutant alleles. The second is c.312_c.321del10insAT in exon 4, which accounted for about 17% of all mutations commonly found in Western, Eastern, and Northern Europeans. The third is c.C688T, which is most commonly found in Northern Europeans. We found c.653_c.654delCT in two reports [41, 49] suggesting that this mutation probably originated in East Asia. One study [50] of 124 cases of LS in China showed that only 4.8% of their patients had mitochondrial DNA mutations, but approximately 20.2% had SURF1 mutations: c.G604C and c.653_c.654delCT. We concluded that different ethnicities have different mutations. It seems that c.845_c.846delCT is typical for people of Slovenian descent as a founder effect [43].”

We have added new references.

  1. Ruhoy IS, Saneto RP: The genetics of Leigh syndrome and its implications for clinical practice and risk management. The application of clinical genetics 2014, 7:221-234.
  2. Gerards M, Sallevelt SC, Smeets HJ: Leigh syndrome: Resolving the clinical and genetic heterogeneity paves the way for treatment options. Molecular genetics and metabolism 2016, 117(3):300-312.

  1. Sofou K, de Coo IFM, Ostergaard E, Isohanni P, Naess K, De Meirleir L, Tzoulis C, Uusimaa J, Lönnqvist T, Bindoff LA et al: Phenotype-genotype correlations in Leigh syndrome: new insights from a multicentre study of 96 patients. Journal of medical genetics 2018, 55(1):21-27.
  2. Ogawa E, Fushimi T, Ogawa-Tominaga M, Shimura M, Tajika M, Ichimoto K, Matsunaga A, Tsuruoka T, Ishige M, Fuchigami T et al: Mortality of Japanese patients with Leigh syndrome: Effects of age at onset and genetic diagnosis. Journal of inherited metabolic disease 2020, 43(4):819-826.
  3. Line 186: Correlation of phenotype and genotype should be further detailed, especially the last part of the paragraph “We do not suggest that outcomes can be predicted based on the results of genetic studies and studies on cytochrome c activity in patients with LS and SURF1 mutations. Possible compensation mechanisms in patients with SURF1 mutations are worthy of further study.” Which mechanisms do the authors mean?. Probably, with the application of machine learning techniques, especially when NGS massive data are available, could in part elucidate this correlation.

Reply: We have rewritten the sentence in the section, and changed to:

“11. Correlation of phenotype and genotype

No clear associations between genotype and phenotype were identified in the clinical features, biochemical analyses and MRI findings of the mutations. This agrees with other studies, [19, 20, 25] and it is difficult to predict LS with SURF1 mutations by genotype because patients with the same mutation/s will often have extremely different expressions. Individual clinical features, such as accelerated neuroregression, should be an important but unpredictable prognostic factor. MRI is also an important, but not an absolute, prognostic factor. In a study from Turkey, 16 cases from 14 families were reported to harbor a recurrent point mutation, c.769G>A [52].The author used three-dimensional (3D) structure prediction for two novel missense variants (C.595_597delGGA and c.356C > T), which cause protein degradation and are predicted as likely pathogenic based on conservation. In this case series, the outcome was poor in all cases, except for one patient with compound heterozygous mutations of c.595_597delGGA (p.Gly199del) and c.751+1 G>A who survived until 18 years of age. However, the reason for this remains unclear. In this case series, no epilepsy was reported, but there were more movement disorders (dystonia and ataxia). Patients with hypotonicity showed shorter survival compared to patients without hypotonicity. However, genotype and phenotype were not related [52]. In one study, patients with SURF1 mutations had worse outcomes of LS compared to those in patients with LS caused by mt-ATP6, NDUF, SLC19A3, and SUCLA2  mutations [47]. For LS caused by SURF1 mutations, the outcomes cannot be predicted based on the results of genetic studies and studies on cytochrome c activity. The possible compensatory mechanisms in patients with SURF1 mutations are worthy of further study. In a previous study, the COX enzyme assembled in the absence of Shy1p (the yeast homologue of SURF1) appears to be structurally and enzymically normal. In vitro labelling studies additionally indicated that mitochondrial translation is significantly increased in the shy1 null mutant strain, reflecting a compensatory mechanism for reduced respiratory capacity [18]. Increased mitochondrial copy number could be a mechanism of compensation and suggests a new direction for mitochondrial biogenesis [53].”

We added the new references

  1. Sofou K, de Coo IFM, Ostergaard E, Isohanni P, Naess K, De Meirleir L, Tzoulis C, Uusimaa J, Lönnqvist T, Bindoff LA et al: Phenotype-genotype correlations in Leigh syndrome: new insights from a multicentre study of 96 patients. Journal of medical genetics 2018, 55(1):21-27.
  2. Kose M, Canda E, Kagnici M, Aykut A, Adebali O, Durmaz A, Bircan A, Diniz G, Eraslan C, Kose E et al: SURF1 related Leigh syndrome: Clinical and molecular findings of 16 patients from Turkey. Molecular genetics and metabolism reports 2020, 25:100657.
  3. Popov LD: Mitochondrial biogenesis: An update. Journal of cellular and molecular medicine 2020, 24(9):4892-4899.
  4. Conclusions (line 337): Authors state that “Coenzyme Q and vitamin cocktails can be supplied early, despite the relatively poor outcomes”. Owing the lack of evidence regarding this issue, I would conclude that it is mandatory to investigate new therapeutic approaches (improve mitochondrial biogenesis, for example, and others) instead to maintain the classical in general non.-effective treatments).

Reply:  In 15. Conclusions, second paragraph, changed to:

 “When patients present with Leigh syndrome or leukodystrophy, SURF1 should be considered. Coenzyme Q and vitamin cocktails can be supplied early, despite the relatively poor outcomes. It is essential to investigate new therapeutic approaches, such as improving mitochondrial biogenesis, to complement the classical treatments that generally have low efficacy. The development of gene replacement therapy has potential in clinical practice for treating LS with SURF1 mutations in the future.”

We added a section “13.5 Mitochondrial biogenesis and gene replacement”

13.5 Mitochondrial biogenesis and gene replacement therapy

In LS with SURF1 mutations, mitochondrial biogenesis is impaired; thus, augmenting mitochondrial biogenesis could be a potential therapeutic approach. Recent reports indicate melatonin promotes mitochondrial biogenesis in the treatment of Alzheimer's disease and Parkinson’s disease [53, 85], and could be a potential drug owing to its influence on mitochondrial physiology. Gene therapy for spinal muscle atrophy (SMA) has recently been developed and has shown promising effects [86]. In LS with SURF1 mutations, intrathecal delivery of adeno-associated viral vector serotype 9 (AAV9)/ human SURF1 (hSURF1) was studied in mice and was found effective in improving the biochemical abnormalities induced by SURF1 deficiency, thus showing potential applicability for patients with SURF1-related Leigh syndrome in the future [87].

We added 3 references

  1. Song C, Li M, Xu L, Shen Y, Yang H, Ding M, Liu X, Xie Z: Mitochondrial biogenesis mediated by melatonin in an APPswe/PS1dE9 transgenic mice model. Neuroreport 2018, 29(18):1517-1524.
  2. Day JW, Finkel RS, Chiriboga CA, Connolly AM, Crawford TO, Darras BT, Iannaccone ST, Kuntz NL, Peña LDM, Shieh PB et al: Onasemnogene abeparvovec gene therapy for symptomatic infantile-onset spinal muscular atrophy in patients with two copies of SMN2 (STR1VE): an open-label, single-arm, multicentre, phase 3 trial. The Lancet Neurology 2021, 20(4):284-293.
  3. Ling Q, Rioux M, Hu Y, Lee M, Gray SJ: Adeno-associated viral vector serotype 9-based gene replacement therapy for SURF1-related Leigh syndrome. Molecular therapy Methods & clinical development 2021, 23:158-168.

Reviewer 2 Report

The review provides a comprehensive summary of the clinical aspects of the SURF1-related Leigh syndrome. The manuscript highlights recent advances in the study of this rare and lethal disease.

I have a few minor comments:

1) There are formatting issues in tables 1 and 2.

2) It would be beneficial to provide a more comprehensive overview of the genetic background of Leigh disease (chapter 1).

3)  Manuscripts gives a comprehensive overview of new development in studying the SURF1 related Leigh disease. Authors might consider discussing a recent publication (2020) of 16 more patients with SURF1 Leigh disease from Turkey (PMID: 33134083).

Author Response

Reviewer 2

We are grateful for the opportunity to improve our manuscript and we thank the editorial board and the reviewers for their thoughtful and helpful comments and criticisms. We have modified the paper as suggested. Following are our point-by-point responses. We have also highlighted the principal changes in the revised text

The review provides a comprehensive summary of the clinical aspects of the SURF1-related Leigh syndrome. The manuscript highlights recent advances in the study of this rare and lethal disease.

I have a few minor comments:

  • There are formatting issues in tables 1 and 2.

Reply: We have formatted in Table 1 and Table 2, including the nomenclature of mutations

  • It would be beneficial to provide a more comprehensive overview of the genetic background of Leigh disease (chapter 1).

Reply: The comprehensive genetic background was added. In 1. Genetic background of Leigh disease, changed to:

  1. Genetic background of Leigh disease

Leigh disease, also called Leigh syndrome (LS), is a genetically heterogeneous disease and can be 1) maternally inherited through mutations in mitochondrial DNA (mtDNA) encoding complex I (MTND1, MTND2, MTND3, MTND4, MTND5, and MTND6), complex IV (MTCO3), complex V (MTATP6) and mitochondrial translation (MTTI, MTTK, MTTL1, MTTV, and MTTW); 2) sex-linked PDHA1 causing pyruvate dehydrogenase deficiency; or 3) autosomal recessive due to mutations in nuclear-encoded complex subunits complex (SDHA and SDHAF1 in complex II deficiency; UQCRQ, BCS1L, and TTC19 in complex III deficiency; and NDUFA4, SURF1, COX10, COX15, SCO2, PET100, LRPPRC, TACO1, and ETHE1 in complex IV deficiency), autosomal recessive in nuclear genes (PDHB, PDHX, DLAT, DLD, LIPT1, LIAS, TPK1,SLC19A3, and SLC25A19) causing pyruvate dehydrogenase deficiency [1, 2], and autosomal recessive in complex assembly genes [3-6]. To date, more than 75 causative genes have been identified to be involved in the biochemical pathways in LS [3-7].

  • Manuscripts gives a comprehensive overview of new development in studying the SURF1 related Leigh disease. Authors might consider discussing a recent publication (2020) of 16 more patients with SURF1 Leigh disease from Turkey (PMID: 33134083).

Reply: We have cited the reference and discuss it.

In 11. Correlation of phenotype and genotype, changed to:

“11. Correlation of phenotype and genotype

No clear associations between genotype and phenotype were identified in the clinical features, biochemical analyses and MRI findings of the mutations. This agrees with other studies, [19, 20, 25] and it is difficult to predict LS with SURF1 mutations by genotype because patients with the same mutation/s will often have extremely different expressions. Individual clinical features, such as accelerated neuroregression, should be an important but unpredictable prognostic factor. MRI is also an important, but not an absolute, prognostic factor. In a study from Turkey, 16 cases from 14 families were reported to harbor a recurrent point mutation, c.769G>A [52].The author used three-dimensional (3D) structure prediction for two novel missense variants (C.595_597delGGA and c.356C > T), which cause protein degradation and are predicted as likely pathogenic based on conservation. In this case series, the outcome was poor in all cases, except for one patient with compound heterozygous mutations of c.595_597delGGA (p.Gly199del) and c.751+1 G>A who survived until 18 years of age. However, the reason for this remains unclear. In this case series, no epilepsy was reported, but there were more movement disorders (dystonia and ataxia). Patients with hypotonicity showed shorter survival compared to patients without hypotonicity. However, genotype and phenotype were not related [52]. In one study, patients with SURF1 mutations had worse outcomes of LS compared to those in patients with LS caused by mt-ATP6, NDUF, SLC19A3, and SUCLA2 mutations [47]. For LS caused by SURF1 mutations, the outcomes cannot be predicted based on the results of genetic studies and studies on cytochrome c activity. The possible compensatory mechanisms in patients with SURF1 mutations are worthy of further study. In a previous study, the COX enzyme assembled in the absence of Shy1p (the yeast homologue of SURF1) appears to be structurally and enzymically normal. In vitro labelling studies additionally indicated that mitochondrial translation is significantly increased in the shy1 null mutant strain, reflecting a compensatory mechanism for reduced respiratory capacity [18]. Increased mitochondrial copy number could be a mechanism of compensation and suggests a new direction for mitochondrial biogenesis [53].”

In 10. Missense and nonsense distributions, changed to:

“10. Missense and nonsense distributions

Based on the missense and nonsense mutations, the ratio of nonsense to missense mutations was 15: 11 (Figure 1). It is thus postulated [41, 51] that nonsense mutations contribute to more severely nonfunctional proteins that cause a more severe phenotype. Among SURF1 mutations, the majority were splice-site, frameshift, and nonsense mutations (Figure 1). Most SURF1 mutations occur in exons 6, 7, and 8, which account for approximately half of the total mutations. Strangely enough, the length of exons 6 to 8 accounts for only a small proportion of the total SURF1 gene. The longest exon, exon 5, contains 192 base pairs, but harbors a small number of different mutations (Figure 1).”

We added the new references

  1. Sofou K, de Coo IFM, Ostergaard E, Isohanni P, Naess K, De Meirleir L, Tzoulis C, Uusimaa J, Lönnqvist T, Bindoff LA et al: Phenotype-genotype correlations in Leigh syndrome: new insights from a multicentre study of 96 patients. Journal of medical genetics 2018, 55(1):21-27.
  2. Kose M, Canda E, Kagnici M, Aykut A, Adebali O, Durmaz A, Bircan A, Diniz G, Eraslan C, Kose E et al: SURF1 related Leigh syndrome: Clinical and molecular findings of 16 patients from Turkey. Molecular genetics and metabolism reports 2020, 25:100657.
  3. Popov LD: Mitochondrial biogenesis: An update. Journal of cellular and molecular medicine 2020, 24(9):4892-4899.

Round 2

Reviewer 1 Report

Authors properly addressed my criticisms.